# Analysis of the Dimensions of Quality of Life in Colombian University Students: Structural Equation Analysis

**DOI:** 10.3390/ijerph17103578

**Published:** 2020-05-20

**Authors:** Juan-Cancio Arcila-Arango, Manuel Castro-Sánchez, Sebastian Espoz-Lazo, Cristian Cofre-Bolados, Maria Luisa Zagalaz-Sánchez, Pedro Valdivia-Moral

**Affiliations:** 1Politécnico Colombiano Jaime Isaza Cadavid, Carrera50D#77Sur81, Medellín 050011, Colombia; jcarcilaa@elpoli.edu.co; 2Department of Didactics of Musical, Plastic and Corporal Expression, University of Granada, 18071 Granada, Spain; pvaldivia@ugr.es; 3Escuela de Salud, Área de Actividad Física, Duoc UC, Santiago de Chile 7510412, Chile; s.espoz.l@gmail.com; 4Laboratory of Sciences of Physical Activity, Sport and Health, Faculty of Medical Sciences, Universidad de Santiago de Chile, Santiago 9170022, Chile; Cristian.cofre@usach.cl; 5Department of Didactics of Musical, Plastic and Corporal Expression, University of Jaén, 23071 Jaén, Spain; lzagalaz@ujaen.es

**Keywords:** motivational climate, Mediterranean diet, body mass index, self-concept

## Abstract

The aim of the present study was to define and contrast a explicative model of the relationship between the variables of quality of life that make up the KIDSCREEN-52 questionnaire. Methods: A total of 1641 Colombian university students aged between 17 and 18 years (17.69 + 0.490) participated in this research (61.2% males and 38.8% females) analyzing the dimensions of the KIDSCREEN-52 quality of life questionnaire. A model of structural equation was made and adjusted (χ^2^ = 118.021; DF = 6; *p* < 0.001; CFI = 0.953; NFI = 0.951; IFI = 0.954; RMSEA = 0.076). Results: The analyzed dimensions of quality of life were related in a positive and direct way, except for the Parent Relationship and Family Life (Family L.) with Social Acceptance (Social A.), which were associated in a negative and indirect manner. Conclusions: The main conclusion of this investigation is that all dimensions of quality of life associate in a positive manner with the exception of Parent Relationship and Family Life (Family L.) which associated with Social Acceptance (Social A.). The qualities improve together, highlighting the idea that working on any of the areas that comprise quality of life will cause development of the remaining areas.

## 1. Introduction

The concept of quality of life has been traditionally related to a correct state of physical health, although in the present this concept has been extended directly to concepts such as satisfaction with the body, happiness, or wellbeing. The development of this concept has gained interest from different disciplines, mainly in the health sciences and social sciences [1,2].

In the Greek era, quality of life was related to a happiness state, while in the 20th century this concept was broadened when different areas such as economy and sociology started to investigate the area, with the School of Chicago beginning studies on social indicators related to quality of life [3,4].

Nowadays, quality of life related to health is mainly characterized by physical, psychological and social wellbeing, positive perceptions about the self, and methods for confronting diseases and adversities [5].

The World Health Organization (WHO) considers quality of life as a concept that depends fundamentally on the physical and mental health of an individual, as well as the quality of their social relationships, the grade of their physical and emotional dependency, personal beliefs, and their integration into social groups. For that, the WHO introduced the following dimensions to define quality of life: physical dimension, psychological dimension, level of dependency, social relationships, environment, and spirituality [6].

Subsequently, the KIDSCREEN group built on this concept and proposed that quality of life is comprised of ten dimensions: Physical Wellbeing, Psychological Wellbeing, Mood and Emotions, Self-Perception, Autonomy, Parent Relationship and Family Life, Economic Resources, Social Support and Peer Relationship, Academic/Professional Environment, and Social Acceptance [7]. The questionnaire “Screening for and Promotion of Health-Related Quality of Life in Children and Adolescent (KIDSCREEN)” is one of the most used instruments for measuring the quality of life related to health in children and adolescents, because they have been adapted to different types of populations and validated in different countries. There are three versions of the KIDSCREEN instrument, one consisting of 10 items (KIDSCREEN-10), another consisting of 27 items (KIDSCREEN-27) and one consisting of 52 items (KIDSCREEN-52). In the present investigation, the KIDSCREEN-52 questionnaire was used, which is made up of 52 items by means of which the subjective health and wellbeing of the subjects is evaluated. The items are valued using a five-point Likert scale. The mean scores obtained in each of the dimensions are standardized to a mean of 50 and a standard deviation of 10, as indicated by the instrument’s creators.

In addition, there are some factors that influence a person’s quality of life such as demographic issues, age, culture, diseases and their treatment centered on the general health of the subject, as well as psychosocial, psychopathological and residency aspects [8].

Nowadays, there is a growing interest in quality of life evaluations across different life stages, motivated mainly by diseases that affect specific age groups, including chronic diseases and, above all, the emotional states that they experience [9]. There have been different investigations that analyze health-related quality of life, such as one by Duran and Castillo [10] that analyzed the differences between the quality of life in university students in different years of university entry. A study by Flores et al. [11] analyzed the relationship between health levels and the quality of life in Mexican students at a public university in Mexico. More recent investigations have also analyzed the quality of life of university students, such as that of Yasartürk et al. [12] that analyzed the quality of life in university students and the satisfaction with the time they dedicate to leisure, or the research by Holt et al. [13] that analyzed the quality of life related to health in university students and its relationship with the use of green spaces.

Time in university is important when quality of life is analyzed because in this part of life subjects experience diverse changes. After adolescence, individuals that begin their university studies confront new situations such as a different academic field compared to their past experiences and a higher state of independency, especially for those who must leave their family home, whereby the social and family relationships also change and people acquire a higher grade of independency in most cases.

Furthermore, these individuals confront a very proximal stage of labor inception, for which stress levels and anxiety tend to rise [14,15]. Health and quality of life of the young is a key factor for all populations due to the importance that these subjects have in terms of social, economic and political development. For these reasons, it is considered fundamental to analyze the different factors of quality of life and how these relate to a university student population, with the objective of being able to identify how they associate between themselves and, in the future, through different investigations, to establish programs centered on the development and improvement of quality of life [16]. Due to the aforementioned information the aims of the present study are to:Define and contrast an explicative model about the relationship between variables that make up quality of life and those which comprise the KIDSCREEN-52 questionnaire.Analyze the existing associations between variables of quality of life and those from the KIDSCREEN-52 questionnaire through a structural equation analysis.

## 2. Materials and Methods 

### 2.1. Design and Participants 

The present descriptive study with transversal cut involved a total sample of 1641 Colombian university students belonging to both genders (61.2% males and 38.8% females), aged between 17 and 18 years old (17.60 + 0.490). Regarding the participant selection process, for this study a sampling by convenience method was applied, following these inclusion and exclusion criteria:Inclusion criteria:
○Aged between 17 and 18 years old.
○Not suffering from diseases or pathology that prevent participation in the investigation.Inclusion criteria:
○If underage, not receiving informed consent from parents or legal tutors.
○Suffering from any disease or pathology that prevents participation in the investigation.

The sample was obtained from the departments of Physical Education, Recreation and Sports; Engineering; Administration; Agrarian Sciences, Basic Sciences; Social and Human and Audio-Visual Communication, requesting the participation of all faculties that voluntarily accepted it.

In the case of underage subjects, informed consent was given by their parents or legal tutors, informing them of the nature of the investigation in order to accept their pupil’s participation. It is important to indicate that the process of data collection ensures no subject repetition in order to avoid duplicating.

### 2.2. Variables and Instrument

The original questionnaire from Ravens-Sieberer et al. [17], “Screening for and Promotion of Health-Related Quality of Life in Children and Adolescent (KIDSCREEN)”, has been selected for this study. It is the most used questionnaire to assess the quality of life related to child and adolescent health, as it has been adapted to different populations in several countries. 

There are three versions of the KIDSCREEN instrument: one composed of 10 items (KIDSCREEN-10), another of 27 items (KIDSCREEN-27) and the last composed of 52 items (KIDSCREEN-52). The latter was used in this investigation in which each item evaluates health and the subjective wellbeing of subjects. The items are measured using a 5-point Likert scale, and has a decafactorial structure with 10 dimensions:○Physical Wellbeing (5 items)○Psychological Wellbeing (6 items)○Mood and Emotions (7 items)○Autonomy (5 items)○Self-Perception (5 items)○Parent Relationship and Family Life (6 items)○Friends and Social Support (6 items)○Scholar Environment (6 items)○Social Acceptance (3 items)○Economic Resources (3 items)

From the surveyed subjects, answers for each mean punctuation were calculated for each dimension to posteriorly standardize them to a mean of 50 with a standard deviation of 10, following the author’s recommendations.

In the studies by Ravens-Sieberer et al. [17], Cronbach’s alpha (α) reliability was determined to be between 0.77 and 0.89 in all ten dimensions [18]. In the present research, the following Cronbach’s alpha scores were obtained: Physical Wellbeing = 0.720; Psychological Wellbeing = 0.870; Mood and Emotions = 0.872; Self-Perception = 0.712; Autonomy = 0.859; Parent Relationship and Family Life = 0.880; Economic Resources = 0.866; Friends and Social Support = 0.844; Scholar Environment = 0.797; Social Acceptance = 0.701.

### 2.3. Procedures 

This section describes the different tasks performed during the data collection field work. The process started by solicitating collaboration from the educational centers selected by convenience. An explicative letter was sent asking for the centers’ and students’ collaboration. In the case of those who were underage, permission from their parent or legal tutor was requested.

After the educational centers accepted, the responsible researcher of the investigation got in contact with the university departments to consent and coordinate the dates for data collection. In the case of underage subjects, informed consent signed by parents or legal guardians was used so that minors could participate in the research.

The current study was conducted according to the Helsinki Declaration (2008 modification) in research projects, respecting the national legislation for clinical trials (223/2004 Law from February 6th), biomedical research (14/2007 Law from July 3rd) and participant’s confidentiality (15/1999 from December 13th).

Regarding the development of the field work, it allowed us to use the questionnaires to collect data from selected adolescents. Finally, is important to indicate that the confidentiality of data and the name of all participants were always respected.

Researchers were present the entire time during data collection in order to avoid any uncertainties on the part of the students. Data collection was developed with no inconvenience and with total normality. It is important to indicate that 116 questionnaires were eliminated due to incompletion.

### 2.4. Data Analysis 

IBM SPSS^®^ (SPSS Inc., Chicago, IL, USA) version 22.0 for Windows^®^ was used in order to perform basic descriptive analysis. IBM AMOS^®^23 (SPSS Inc., Chicago, IL, USA) was used to analyze the existing relationships between implied constructs from the structural model.

A general model was tested in which all scales and sub-scales used in this research were included in order to analyze the interactions between the predictions of the resulting variables. Finally, a theoretical model with the best indexes of adjustment goodness was chosen due to its response to the bibliographic reference and its capacity to explain the achieved results from the investigation.

Once the theoretical model was developed, a route analysis was performed, considering the matrix relationships regarding a structural equation analysis.

The routes models are composed of five observable variables and two latent variables to determine the indicators (Figure 1). In these, causal explanations of the latent variables were formulated regarding the observed variables between indicators, considering the reliability of measures. Likewise, errors of observable variable measurements were included in order to control them directly. Unidirectional arrows are lines of influence between latent and observable variables, and these were interpreted as multivariate regression coefficient. Bidirectional arrows indicate the relationship between latent variables, also representing the regression coefficient.

Physical Wellbeing (Physical W.) and Psychological Wellbeing (Psychological W.) act as exogenous variables. Mood and Emotions (Mood E.) acts as an endogenous variable receiving the effect of Physical Wellbeing (Physical W.), Psychological Wellbeing (Psychological W.), and Self-Perception. Self-Perception acts as an endogenous variable receiving the effect of Physical Wellbeing (Physical W.) and Psychological Wellbeing (Psychological W.). Autonomy acts as an endogenous variable receiving the effect of Physical Wellbeing (Physical W.), Psychological Wellbeing (Psychological W.), and Self-Perception. The Relationship with parents and family life (Family L.) and Social Acceptance (Social A.) act as endogenous variables receiving the effect of Mood and Emotions (Mood E.), Self-Perception, and Autonomy.

The model adjustment was checked in order to verify its compatibility with itself and with the obtained empirical information. The reliability of the adjustment was performed according to Marsh’s criteria of adjustment goodness [19].

In the case of chi-square, not significant values associated with *p* indicate a good adjustment of the model. The comparative adjustment index (CFI) was acceptable with values over 0.90 and excellent if values were higher than 0.95. The normalized adjustment index value (NFI) had to be higher than 0.90. The value of the adjustment increase index (IFI) was acceptable with values greater than 0.90 and excellent for values greater than 0.95. Finally, the value of the mean square error of approximation (RMSEA) was excellent if it was under 0.05 and acceptable if it was lower than 0.08.

## 3. Results

The proposed structural equation model reveals a good adjustment in all evaluation indexes. Chi-square shows a significant *p* value (χ^2^ = 118.021; DF = 6; *p* < 0.001). However, this index cannot be interpreted in a standardized manner, and shows sensibility to sample size [15]. In this way, less sensitive standardized adjustments indexes were used for sample size. CFI obtained a value of 0.953, which was considered excellent. The NFI was 0.951 and the IFI was 0.954, both excellent. RMSEA had an acceptable value of 0.076. 

In Figure 2 and Table 1 the estimated values of the structural model parameters are shown. These must present an adequate magnitude and the effects must be a significant zero. Likewise, improper estimations as negative variances must not be obtained.

A positive and direct association is observed at the *p* < 0.005 level between Physical Wellbeing and Psychological Wellbeing (r = 0.423). Psychological Wellbeing is associated positively and directly at the *p* < 0.005 level with Mood and Emotions (r = 0.417), Self-Perception (r = 0.375), and Autonomy (r = 0.287). Physical Wellbeing is associated in a positive and direct way at the *p* < 0.005 level with Autonomy (r = 0.210), and at the <0.05 level with Mood and Emotions (r = 0.069) and Self-Perception (r = 0.072). Mood and Emotions is associated positively and directly at the *p* < 0.005 level with Parent Relationship and Family Life (r = 0.282) and Social Acceptance (r = 0.274). Self-Perception is positively and directly associated at the *p* < 0.005 level with Autonomy (r = 0.117), Mood and Emotions (r = 0.222), Parent Relationship and Family Life (r = 0.130), and Social Acceptance (r = 0.149). Autonomy is associated positively and directly at the *p* < 0.005 level with Parent Relationship and Family Life (r = 0.186), and negatively and indirectly at the p < 0.05 level with Social Acceptance (r = −0.062).

## 4. Discussion

The present investigation carried out an analysis of structural equations, with the objective of contrasting the associations between the different dimensions of quality of life used in the KIDSCREEN-52 questionnaire. The developed route model acquired good adjustment indexes, configuring a valid explanatory model that allows explanation of the associations between the different dimensions of quality of life, as has been conducted in various studies [20,21,22,23,24].

Based on the relationship between Physical Wellbeing and Psychological Wellbeing, a positive and direct association was found. These data coincide with those found in research by Gaspar et al. [25], in which the association between the different dimensions that make up the health-related quality of life in subjects from Portugal was analyzed through correlations, with the finding that there was also a positive association between both dimensions. Physical Wellbeing is usually related to proper nutrition and the practice of regular physical activity, which ultimately leads to various health benefits, including a reduction in stress, increased motivation, the physical dimension of self-concept, increased self-esteem, and high levels of tolerance for frustration [26,27,28]. Therefore, the association between Physical and Psychological Wellbeing seems clear since when one increases its benefits affect the other dimensions [29,30,31].

When analyzing the relationship between Physical and Psychological Wellbeing with Mood and Emotions, Self-Perception, and Autonomy, there is a positive and direct association. These data suggest the idea that physical and cognitive wellbeing can affect other wellbeing dimensions, influencing them positively [32]. When Physical and Psychological Wellbeing increase, then Mood and Emotions, Self-Perception, and Autonomy do in the same way, with the understanding that these types of wellbeing affect the rest and the dimensions are positively related [33].

In addition, a direct and positive association was found between Mood and Emotions with Parent Relationship and Family Life, and Social Acceptance. These data can be explained according to the idea that when a subject has good social relations and relationships with parents and family are positive, their mood and emotional health will also be affected positively [34,35,36]. Furthermore, when subjects enjoy a good mood and good emotional health, relationships with their parents and peer groups will be more fruitful. In some research it has been found that dysfunctionalities in social and family relationships greatly affect individuals, causing various behavioral and emotional problems [37,38,39].

In view of Self-Perception, there is a positive and direct association with Autonomy, Mood and Emotions, Parent Relationship and Family Life, and Social Acceptance. This highlights the importance of the perception that the subjects have of themselves, because when individuals perceive themselves correctly, they increase their self-concept and self-esteem [40,41,42]. These results can be explained because when a subject has a correct perception of themselves, by increasing their self-concept and self-esteem there is an improvement in their social relations, both with family and with their peer group [43,44,45].

Finally, there is a positive and direct association between Autonomy and Parent Relationship and Family Life, while this relationship is negative between Autonomy and Social Acceptance. The autonomy that family offers to subjects causes fruitful family relationships, because a subject with greater autonomy will be perceived as more capable and will feel part of the family nucleus, therefore reinforcing these relationships [46,47,48]. However, in the case of the association between Autonomy and Social Relations with the peer group, the association is negative because people who enjoy great autonomy usually appreciate the time they spend alone, which does not mean that they are less social people. However, when a person enjoys their time alone, they usually dedicate less time to social relations, which could explain the association [49,50,51,52].

In the existing literature, some research has analyzed the association between the different dimensions that make up health-related quality of life, but no research has analyzed the association between the different dimensions that make up a structural equation model in the health-related quality of life on the KIDSCREEN-52 questionnaire. For these reasons, this research provides interesting data on the associations between different dimensions of quality of life that occur in a sample of students from Colombia. It would be interesting in the future to conduct similar research in different populations to compare the obtained data.

## 5. Conclusions

The main conclusion of the present investigation is that there is a positive and direct association between Physical Wellbeing and Psychological Wellbeing. Equally, Psychological Wellbeing and Physical Wellbeing associate directly and positively with Mood and Emotions, Self-Perception, and Autonomy. Mood and Emotions associates positively and directly with Parent Relationship and Family Life and Social Acceptance.

From these conclusions it is understood that all the dimensions of quality of life are positively associated with each other, except for Parent Relationship and Family Life with Social Acceptance. The qualities strengthen each other and highlighted from this is the idea that working on any of the areas that make up quality of life will cause developments in the rest of the areas. As practical implications of this research, we suggest that the development of Physical and Psychological Wellbeing must be promoted in the first instance in order to develop the rest of the factors related to quality of life and thus improve the general state of health. This study has a series of limitations, among which is the design, since it is descriptive and has a cross-sectional nature which does not allow establishment of cause–effect relationships. In addition, it would have been interesting to include a greater number of psychological variables in order to analyze their relationship with the factors studied in the investigation and verify how they relate to quality of life.

The main novelty of this research lies in the analysis of the association between the different dimensions of the quality of life questionnaire (Kidscreen-52) in a population of adolescents in Colombia.

## Figures and Tables

**Figure 1 ijerph-17-03578-f001:**
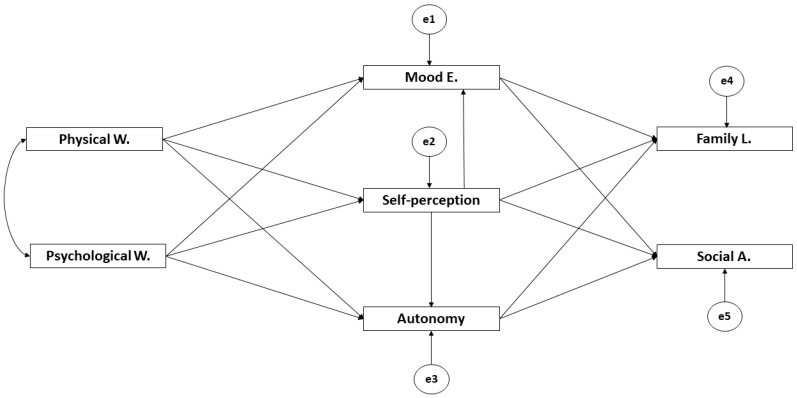
Theoretical model.

**Figure 2 ijerph-17-03578-f002:**
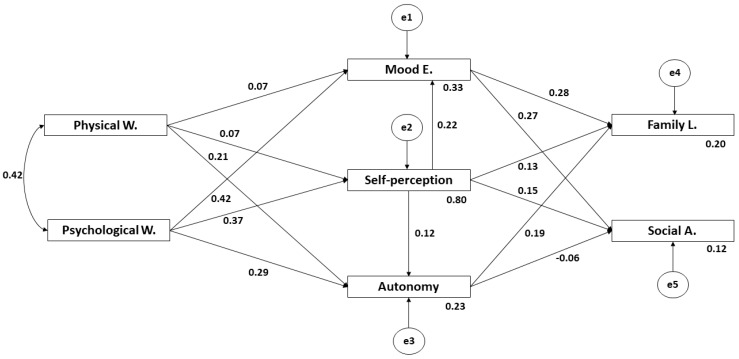
Structural equation model.

**Table 1 ijerph-17-03578-t001:** Structural model.

Relationship between Variables			R.W.				S.R.W.
			Estimations	E.E.	C.R.	*p*	Estimations
Self-Perception	←	Physical W.	0.055	0.019	2904	*	0.072
Self-Perception	←	Psychological W.	0.295	0.020	15.071	***	0.375
Mood E.	←	Physical W.	0.060	0.019	3092	*	0.069
Autonomy	←	Physical W.	0.194	0.022	8743	***	0.210
Autonomy	←	Psychological W.	0.272	0.024	11.238	***	0.287
Mood E.	←	Psychological W.	0.373	0.021	17.548	***	0.417
Autonomy	←	Self-perception	0.141	0.029	4932	***	0.117
Mood E.	←	Self-perception	0.253	0.025	10.049	***	0.222
Family L.	←	Mood E.	0.334	0.029	11.500	***	0.282
Family L.	←	Self-Perception	0.175	0.033	5275	***	0.130
Social A.	←	Self-Perception	0.217	0.038	5759	***	0.149
Family L.	←	Autonomy	0.208	0.026	7960	***	0.186
Social A.	←	Autonomy	−0.075	0.030	−2.532	*	−0.062
Social A.	←	Mood E.	0.350	0.033	10.641	***	0.274
Physical W.		Psychological W.	33.434	2.117	15.792	***	0.423

Note1: R.W. Regression weight; S.R.W. Standardized regression weight; E.E. Error estimation; C.R., Critical ratio. Note 2: *** Relationship between variables statistically significant at the 0.005 level; * Relationship between variables statistically significant at the 0.05 level.

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
