# Peer review of "Analysis of the Dimensions of Quality of Life in Colombian University Students: Structural Equation Analysis"

_ijerph, 2020, doi:10.3390/ijerph17103578_

Round 1

Reviewer 1 Report

Comments to authors

It is a great opportunity to review a manuscript, entitled “Analysis of the dimensions of quality of life in Colombian university students: structural equation analysis.” The authors assessed the Colombian University Students’ quality of life by using the 10 dimensions of the KidScreen-52 Questionnaire.

This is an interesting academic paper; however, the authors should address the following concerns:

  • The introduction:
  • It is very generic. Authors focused too much on the general introduction of definitions of quality of life (QoL). The main focus should lie down between what has been done related to QoL of university students in other universities of your country and the rest of the world.
  • Since the authors used the KidScreen-52 questionnaire, it is important to introduce what it is about and how it is measured. What makes your results differ from previous studies that used KidScreen-52 Questionnaire.
  • The objective is not so clear.
  • The method and material
  • The first two paragraphs are very general and less relevant. It is suggested to remove.
  • The inclusion and exclusion criteria are very wide. They came out with some doubts about how you classified your respondents. For example, how did you know that they had suffered diseases or pathology? Did you provide a medical test or they show you the medical report?
  • It is useful to inform readers how and why you selected students from those majors, why not others? How did you reach the students' or parents’ consent? How did you deliver your questionnaire to them? When did the students fill in your ­survey? Was it during their class session?
  • In section 2.2. “Variables and Instrument,” you described 10 dimensions, but why in the model you had only 7 dimensions?
  • Furthermore, on page 5 first paragraph you wrote a variable, i.e., Physical Wellbeing (Physical W.), several times. Why didn’t you use the words in the bracket when you wrote them again? It was very confusing.
  • Results
  • What is the main significance of your results compared with previous studies using the KidScreen-52 Questionnaire?
  • Discussion
  • L231 You mentioned, “…..on the relationship between Physical Wellbeing and Psychological Wellbeing, a positive and direct association has been found.” Didn’t previous studies find such a relationship? If yes, how can your study be different from others?
  • Conclusion
  • What is the novelty of your study?

Author Response

Dear the editor and reviewers,

We would like to express our gratitude for the time taken to review this manuscript and for the comments made, which we believe to be critical for producing rigorous and quality research. We have detailed below the changes made in the original article: “Analysis of the dimensions of quality of life in Colombian university students: structural equation analysis” (ijerph-785174).

Modifications have been made in the original manuscript following the reviewers’ comments. For each modification we have written: the original comment as written by the reviewer in addition to the page and line number; and the change made in response to that comment. Changes have been made using the tool “Track changes” enabling editor and reviewers to identify modifications easily.

Reviewer 1

Comment 1:

It is a great opportunity to review a manuscript, entitled “Analysis of the dimensions of quality of life in Colombian university students: structural equation analysis.” The authors assessed the Colombian University Students’ quality of life by using the 10 dimensions of the KidScreen-52 Questionnaire.

This is an interesting academic paper; however, the authors should address the following concerns:

    The introduction:

    It is very generic. Authors focused too much on the general introduction of definitions of quality of life (QoL). The main focus should lie down between what has been done related to QoL of university students in other universities of your country and the rest of the world.

    Since the authors used the KidScreen-52 questionnaire, it is important to introduce what it is about and how it is measured. What makes your results differ from previous studies that used KidScreen-52 Questionnaire.

    The objective is not so clear.

Response 1:

Dear reviewer, thank you very much for your comments.

As for the introduction, what this model intends is to analyze the relationship between the dimensions of the questionnaire, so there is no literature with which to compare the data, we very much appreciate your comment.

Regarding the objective, it is indicated that it is intended to carry out a predictive model that shows how the associations between the dimensions that make up the Kidscreen-52 questionnaire are produced.

Comment 2:

The method and material

The first two paragraphs are very general and less relevant. It is suggested to remove.

The inclusion and exclusion criteria are very wide. They came out with some doubts about how you classified your respondents. For example, how did you know that they had suffered diseases or pathology? Did you provide a medical test or they show you the medical report?

It is useful to inform readers how and why you selected students from those majors, why not others? How did you reach the students' or parents’ consent? How did you deliver your questionnaire to them? When did the students fill in your ­survey? Was it during their class session?

In section 2.2. “Variables and Instrument,” you described 10 dimensions, but why in the model you had only 7 dimensions?

Furthermore, on page 5 first paragraph you wrote a variable, i.e., Physical Wellbeing (Physical W.), several times. Why didn’t you use the words in the bracket when you wrote them again? It was very confusing.

Response 2:

Thanks for this indication. As for the first two paragraphs of the discussion, they have been removed as indicated by the reviewer.

Regarding the inclusion criteria, they have also been modified eliminating one of them, as indicated by the other reviewer.

Regarding diseases, it was only intended to know if they had any serious pathology that affected the completion of the questionnaire, so the subjects surveyed were asked directly.

Regarding the selection of the sample, this section indicates that a convenience sampling was carried out.

In the case of the minor subjects, consent was passed to the parents and the researchers carried out the data collection process by directly administering the questionnaires to the subjects that make up the sample. This information has been added to the investigation.

Regarding the structural equations model, the data analysis section indicates that since the model in which all the dimensions were included did not provide good adjustments, it was modified to obtain adjustments that would make the model valid.

Regarding the indications referring to the names of the variables, the full name and the abbreviation are mentioned as it appears in the model image, for clarification. This is done to explain what are the endogenous and exogenous variables.

Comment 3:

Results

What is the main significance of your results compared with previous studies using the KidScreen-52 Questionnaire?.

Response 3:

Thanks for this indication. The main novelty of the results that this study shows is that a structural equation model is carried out to analyze the association between the variables that make up the Kidscreen questionnaire and it is checked whether they are analyzed positively or negatively in a population of adolescents in Colombia.

Comment 4:

Discussion

L231 You mentioned, “…..on the relationship between Physical Wellbeing and Psychological Wellbeing, a positive and direct association has been found.” Didn’t previous studies find such a relationship? If yes, how can your study be different from others?

Response 4:

Thank you very much for this suggestion of improvement. No literature has been found in the respect that analyzes this association, so it is indicated why it is believed that this relationship may exist, but it cannot be compared with other studies.

Comment 5:

Conclusion

What is the novelty of your study?.

Response 5:

Thanks for this indication. It has been explained in the conclusions section that the main novelty of this research lies in the analysis of the association between the different dimensions of the quality of life questionnaire (Kidscreen-52) in a population of adolescents in Colombia..

Reviewer 2 Report

- The research topic is original, it is also of interest to the scientific community. The study age is adequate to carry out this study, it is very well justified. First-year university students are strengthening their personality and study variables are important at these ages.
- The data analysis is very good, it is very well explained. As stated in the manuscript, the study could be replicated perfectly.
- The explanation of the limitations of the study is appreciated.

Below I will constructively propose some questions to improve the study.

1. On line 33 the word satisfaction appears. It would be necessary to indicate if it is satisfaction with the body (self-esteem is spoken of in the variables)
2. It would eliminate the inclusion criterion 1. In my opinion it is not necessary.
3. In relation to the discussion, it would be convenient to explain in more detail the negative relationship between autonomy and social acceptance. As well as having greater support from the bibliography.
In all cases, being more autonomous does not mean spending more time alone ... I think this part should be improved.

Author Response

Dear the editor and reviewers,

We would like to express our gratitude for the time taken to review this manuscript and for the comments made, which we believe to be critical for producing rigorous and quality research. We have detailed below the changes made in the original article: “Analysis of the dimensions of quality of life in Colombian university students: structural equation analysis” (ijerph-785174).

Modifications have been made in the original manuscript following the reviewers’ comments. For each modification we have written: the original comment as written by the reviewer in addition to the page and line number; and the change made in response to that comment. Changes have been made using the tool “Track changes” enabling editor and reviewers to identify modifications easily.

RevieweR 2

Comment 1:

- The research topic is original, it is also of interest to the scientific community. The study age is adequate to carry out this study, it is very well justified. First-year university students are strengthening their personality and study variables are important at these ages.

- The data analysis is very good, it is very well explained. As stated in the manuscript, the study could be replicated perfectly.

- The explanation of the limitations of the study is appreciated.

Below I will constructively propose some questions to improve the study.

  1. On line 33 the word satisfaction appears. It would be necessary to indicate if it is satisfaction with the body (self-esteem is spoken of in the variables).

Response 1:

We appreciate your indications as they improve the quality of the article. The indicated modifications have been made, reaping the idea that it talks about satisfaction with the body.

Comment 2:

  1. It would eliminate the inclusion criterion 1. In my opinion it is not necessary.

Response 2:

Estimado revisor, muchas gracias por su revision del manuscrito tan produnda. Agradecemos mucho su labor revisando y corrigiendo todos los errors. Has been removed the inclusion criterion 1.

Comment 3:

  1. In relation to the discussion, it would be convenient to explain in more detail the negative relationship between autonomy and social acceptance. As well as having greater support from the bibliography.

In all cases, being more autonomous does not mean spending more time alone ... I think this part should be improved.

Response 3:

Dear Reviewer, Thank you very much for your thorough review of the manuscript. We greatly appreciate your work reviewing and correcting all errors. When analyzing this association, not much scientific information is found in this regard, but it is intuited that being more autonomous and depending less on others, this can decrease dependency relationships towards others and as a result show lower scores in relationships social.

Round 2

Reviewer 1 Report

Dear authors

I thank for the responses that the authors replied to the comments that I gave in the first version. Unfortunately, I did not see much change. For example, in the introduction, the authors responded, “As for the introduction, what this model intends is to analyze the relationship between the dimensions of the questionnaire, so there is no literature with which to compare the data, we very much appreciate your comment.” Based on this response, I have other questions:

  • What are the main contributions your paper will produce? Only to the KidScreen-52Questionnaire?
  • If yes, I doubt that why previous studies did not investigate the relationship between dimensions of the questionnaire if it is important?
  • If there is no previous study, provide some evidence (e.g., which authors confirmed the relationship between dimensions has not been done? How is this important to find the relationship between dimensions?) to support your assumption.

Likely, in the discussion, the authors responded that “No literature has been found in the respect that analyzes this association,…”

To my understanding, the KidScreen questionnaire has been adopted for several years in different fields of studies, esp. public health. It is strange that previous studies did not measure such a relationship. Ok, let’s assume that there is no literature but you should explain to readers why the previous do not pay attention to the relationship between dimensions of the questionnaire? Isn’t it important to do it? If you can answer these questions, maybe your findings are attractive to the readers who are interested in the Kidscreen-52 Questionnaire.

I do not see anything new, except for the relationship between dimensions.

I may consider your paper again after you can make a second MAJOR revision.

Author Response

Dear the editor and reviewers,

We would like to express our gratitude for the time taken to review this manuscript and for the comments made, which we believe to be critical for producing rigorous and quality research. We have detailed below the changes made in the original article: “Analysis of the dimensions of quality of life in Colombian university students: structural equation analysis” (ijerph-785174).

Modifications have been made in the original manuscript following the reviewers’ comments. For each modification we have written: the original comment as written by the reviewer in addition to the page and line number; and the change made in response to that comment. Changes have been made using the tool “Track changes” enabling editor and reviewers to identify modifications easily.

Reviewer 1

Comment 1:

What are the main contributions your paper will produce? Only to the KidScreen-52Questionnaire?

If yes, I doubt that why previous studies did not investigate the relationship between dimensions of the questionnaire if it is important?.

Response 1:

Dear reviewer, thank you very much for your comments.

Actually, the main contribution of this research is an analysis of the dimensions that make up the quality of life through a structural equation model, in a population of students from Colombia.

Only one research article was found that analyzes four dimensions of quality of life (Psychological, Emotional, Social and Family Relations) using a structural equation model. I include research reference:

Titman, A. C., Lancaster, G. A., & Colver, A. F. (2016). Item response theory and structural equation modelling for ordinal data: Describing the relationship between KIDSCREEN and Life-H. Statistical methods in medical research, 25(5), 1892-1924.

If there is any research that analyzes the relationships between the dimensions, but not using a structural equation model.

Comment 2:

If there is no previous study, provide some evidence (e.g., which authors confirmed the relationship between dimensions has not been done? How is this important to find the relationship between dimensions?) to support your assumption.

Response 2:

Thanks for this indication. Only one investigation has been found that analyzes the relationships between the dimensions of the quality of life questionnaire, therefore, due to the paucity of investigations in this regard, it is considered necessary to increase the number of these. The article found in this relationship is:

Gaspar, T., De Matos, M. G., Batista-Foguet, J., Pais Ribeiro, J. L., & Leal, I. (2010). Parent–child perceptions of quality of life: Implications for health intervention. Journal of Family Studies, 16(2), 143-154.

Comment 3:

Likely, in the discussion, the authors responded that “No literature has been found in the respect that analyzes this association,…”

To my understanding, the KidScreen questionnaire has been adopted for several years in different fields of studies, esp. public health. It is strange that previous studies did not measure such a relationship. Ok, let’s assume that there is no literature but you should explain to readers why the previous do not pay attention to the relationship between dimensions of the questionnaire? Isn’t it important to do it? If you can answer these questions, maybe your findings are attractive to the readers who are interested in the Kidscreen-52 Questionnaire.

Response 3:

Thank you very much for your indications and reflections on the present investigation, we consider that this will improve the quality of the investigation.

Most of the investigations that analyze the quality of life and specifically that use the Kidscreen-52 questionnaire carry out analyzes of the dimensions of the quality of life with other different variables, only the investigation mentioned in the previous comment has been found in which performs a correlation analysis between the dimensions that make up the quality of life.

For these reasons, the modifications indicated in the first revision have been carried out, comparing the data with those shown in this research article:

Comment 1-First Review:

It is a great opportunity to review a manuscript, entitled “Analysis of the dimensions of quality of life in Colombian university students: structural equation analysis.” The authors assessed the Colombian University Students’ quality of life by using the 10 dimensions of the KidScreen-52 Questionnaire.

This is an interesting academic paper; however, the authors should address the following concerns:

    The introduction:

    It is very generic. Authors focused too much on the general introduction of definitions of quality of life (QoL). The main focus should lie down between what has been done related to QoL of university students in other universities of your country and the rest of the world.

    Since the authors used the KidScreen-52 questionnaire, it is important to introduce what it is about and how it is measured. What makes your results differ from previous studies that used KidScreen-52 Questionnaire.

Response 1:

We appreciate your review.

As for the previous studies that analyze the quality of life in different groups, this information has been added to expand the introduction and provide the data that we consider necessary. Research carried out in Spanish-speaking countries and more recent international studies have been added, all of them carried out on populations of university students.

Regarding the description of the instrument, we have proceeded to describe it in a broader way, since the reviewer is absolutely right in recommending that it be specified more, since the present investigation focuses on the analysis of the dimensions used by the instrument. Kidscreen-52 questionnaire.

Comment 2-First Review:

Comment 2 was reviewed and the modifications indicated by the reviewer were made during the first review.

Comment 3-First Review:

Comment 3 was reviewed and the modifications indicated by the reviewer were made during the first review.

Comment 4-First Review:

Discussion

L231 You mentioned, “…..on the relationship between Physical Wellbeing and Psychological Wellbeing, a positive and direct association has been found.” Didn’t previous studies find such a relationship? If yes, how can your study be different from others?

Response 4:

Thank you very much for this suggestion of improvement. Carrying out a more specific search we have found an investigation that analyzes this relationship. For this reason, it has been included in the discussion, indicating that the data obtained coincides with that found in this investigation. The research found that performs the correlation analysis is:

Gaspar, T., De Matos, M. G., Batista-Foguet, J., Pais Ribeiro, J. L., & Leal, I. (2010). Parent–child perceptions of quality of life: Implications for health intervention. Journal of Family Studies, 16(2), 143-154.

Comment 5-First Review:

Comment 5 was reviewed and the modifications indicated by the reviewer were made during the first review.

Round 3

Reviewer 1 Report

Dear Authors,

I can see a significant effort you have made to respond to my questions. It is good to see some changes in the original manuscript. While responding to my questions, it is better that you also add what you responded to the manuscript, especially in the discussion so as to inform the readers that your research has some contribution to the existing literature.

Author Response

Dear the editor and reviewers,

We would like to express our gratitude for the time taken to review this manuscript and for the comments made, which we believe to be critical for producing rigorous and quality research. We have detailed below the changes made in the original article: “Analysis of the dimensions of quality of life in Colombian university students: structural equation analysis” (ijerph-785174).

Modifications have been made in the original manuscript following the reviewers’ comments. For each modification we have written: the original comment as written by the reviewer in addition to the page and line number; and the change made in response to that comment. Changes have been made using the tool “Track changes” enabling editor and reviewers to identify modifications easily.

Reviewer 1

Comment 1:

I can see a significant effort you have made to respond to my questions. It is good to see some changes in the original manuscript. While responding to my questions, it is better that you also add what you responded to the manuscript, especially in the discussion so as to inform the readers that your research has some contribution to the existing literature.

Response 1:

Dear reviewer, thank you very much for your comments. An effort has really been made to respond to all indications and suggestions made. As for the indications to add the answers to your questions in the document to make it clearer that this research contributes as a novelty, this information has been included.

The information requested by the reviewer has been included in the discussion in order to clarify all doubts to the reader and influence the contributions of this research, as well as the need to carry out similar studies in order to collate the data, since There is no scientific literature that analyzes these associations between the dimensions of quality of life in different populations using structural equation models.

Thank you for your feedback on improvement, we believe this will improve the quality of our manuscript.